# Reward Finetuning for Faster and More Accurate Unsupervised Object Discovery

**Katie Z Luo**[*,1,†] **Zhenzhen Liu**[*,1] **Xiangyu Chen**[*,1] **Yurong You**[1] **Sagie Benaim**[2] **Cheng Perng Phoo**[1]
**Mark Campbell**[1] **Wen Sun**[1] **Bharath Hariharan**[1] **Kilian Q. Weinberger**[1]
[1]Cornell University, Ithaca, NY    [2]The Hebrew University of Jerusalem

## Abstract

Recent advances in machine learning have shown that Reinforcement Learning from Human Feedback (RLHF) can improve machine learning models and align them with human preferences. Although very successful for Large Language Models (LLMs), these advancements have not had a comparable impact in research for autonomous vehicles—where alignment with human expectations can be imperative. In this paper, we propose to adapt similar RL-based methods to unsupervised object discovery, i.e. learning to detect objects from LiDAR points without any training labels. Instead of labels, we use simple heuristics to mimic human feedback. More explicitly, we combine multiple heuristics into a simple reward function that positively correlates its score with bounding box accuracy, i.e., boxes containing objects are scored higher than those without. We start from the detector's own predictions to explore the space and reinforce boxes with high rewards through gradient updates. Empirically, we demonstrate that our approach is not only more accurate, but also orders of magnitudes faster to train compared to prior works on object discovery. Code is available at https://github.com/katieluo88/DRIFT.

## 1 Introduction

Self-driving cars need to accurately detect the moving objects around them in order to move safely. Most modern 3D object detectors rely on supervised training from 3D bounding box labels. However, these 3D bounding box labels are hard to acquire from human annotation. Furthermore, this supervised approach relies on a pre-decided vocabulary of classes, which can cause problems when the car encounters novel objects that were never annotated.

Our prior work, MODEST [55], introduced the first method to train 3D detectors without labeled data. In that work, we point out that instead of specifying millions of labels, one can succinctly describe *heuristics* for what a good detector output should look like. For example, one can specify that detector boxes should mostly enclose transient foreground points rather than background ones; they should roughly be of an appropriate size; their sides should be aligned with the LiDAR points; their bottom should touch the ground, etc. Although such heuristics are great for *scoring* a set of boxes proposed by a detector, training a detector on them is hard for two reasons: First, these heuristics are often non-linear, non-differentiable functions of the detector parameters (for example, a slight shift of the box can cause all foreground points to fall off.) Second, existing object detection pipelines use carefully designed training objectives that heavily rely on labeled boxes, that are difficult to modify (for example, PointRCNN [40] infers point labels from box labels and uses these for training). For these reasons, MODEST had to utilize an admittedly slow self-training pipeline to incrementally incorporate common-sense heuristics.

---

[*]Denotes equal contribution.
[†]Correspondences could be directed to kzl6@cornell.edu

37th Conference on Neural Information Processing Systems (NeurIPS 2023).

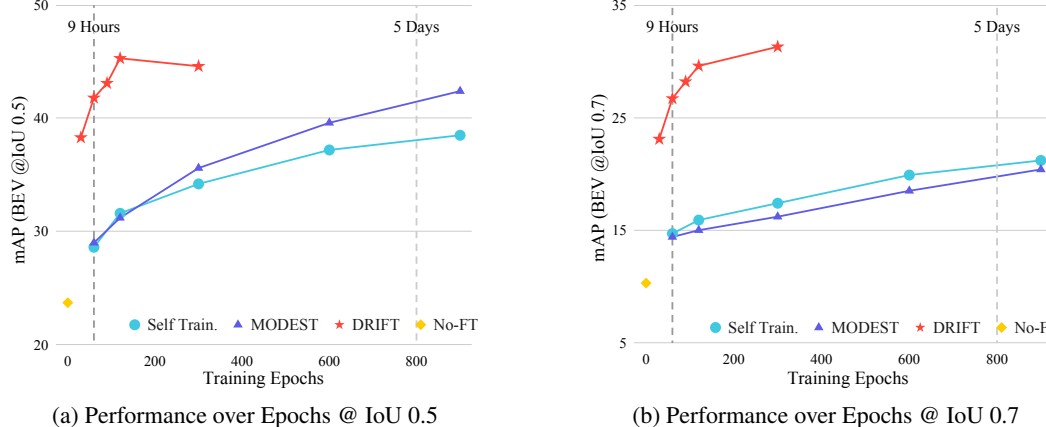

(a) Performance over Epochs @ IoU 0.5      (b) Performance over Epochs @ IoU 0.7

Figure 1: **Detection performance on Lyft test data as a function of training epochs.** DRIFT demonstrates significantly stronger performance and faster learning. With only 9 hours of training, it outperforms both baselines that have been trained for days.

In this paper, we propose a new reward ranking based framework that utilizes these common-sense heuristics directly as a reward signal. Our method relies on finetuning with reward ranking [31, 27, 13, 33], where given an initialized object detector, we finetune it for a predefined set of desirable detection properties. This bypasses the need to encode heuristics as differentiable loss functions and avoids the need to hand-engineer training paradigms for each kind of object detector. Recent success with reinforcement learning from human feedback (RLHF) has proven effective in improving machine learning models —in particular, large language models (LLMs)— and aligning them with human preferences [31, 27]. However, these advancements have not been applicable to detection-based vision models that are trained with per-instance regression and are difficult to view under a probabilistic framework. To address this challenge, we utilize insights from reward ranked finetuning [13], a non-probabilistic paradigm designed for finetuning of LLMs, which inspired us to develop a similar framework for object discovery.

We refer to our method as *Discovery from Reward-Incentivized Finetuning (DRIFT)*. DRIFT does not require labels, and instead uses the Persistency Prior (PP) score [55, 3] as a heuristic to identify dynamic foreground points based on historical traversals. These foreground points give rise to rough (and noisy) label estimates [55], which we use to pre-train our detector. The resulting detector performs poorly but suffices to propose many boxes of roughly the right sizes that we can use for exploration. To facilitate reward ranked finetuning, we first propose a reward function to score boxes. Ideally, only boxes that tightly contain objects (e.g. a car) should yield high rewards. We achieve this by combining several simple heuristics (e.g. high ratio of foreground points) and assuming some rough knowledge about the object dimensions. During each iteration of training, DRIFT performs the following steps: 1. the object detector proposes many boxes in a given point cloud scene; 2. the boxes are "jittered" through random perturbations (as a means of exploration); 3. the boxes are scored according to the reward function; 4. the top-$k\%$ non-overlapping boxes are kept as pseudo-labels for gradient updates.

We evaluate DRIFT on two large, real-world datasets [12, 24] and show that we significantly outperform prior self-training methods both in efficiency and generalizability. Experimental results demonstrate that using reward ranked finetuning for object discovery under our framework can quickly converge to a solution that is on par with out-of-domain supervised object detectors within a few training epochs, suggesting that DRIFT may point towards a more general unsupervised learning formulation for object detectors in an in-the-wild setting.

## 2    Related Works

**3D Object Detection.** 3D object detection models usually take in LiDAR point clouds or multi-view images as input and aim to produce tight bounding boxes that describe nearby objects [9, 52, 26, 39, 40, 53, 54, 34]. Existing methods generally assume the supervised setting, in which the detector is trained with human-annotated bounding boxes. However, annotated data are often expensive to obtain

and limited in quantity. Furthermore, in tasks such as self-driving, environments can have highly varied conditions, and detectors with supervised training often require adaptation with additional labels from the new environment [47].

**Unsupervised Object Discovery**. The unsupervised object discovery task aims to identify and localize salient objects without learning from labels. Most existing works perform discovery from 2D images [8, 14, 45, 36, 2, 43, 48] or depth camera frames [21, 23, 17, 25, 20, 1]. Discovery from 3D LiDAR point clouds is underexplored. [49] performs joint unsupervised 2D detection and 3D instance segmentation from sequential point clouds and images based on temporal, motion and correspondence cues. MODEST [55] pioneers in performing label-free 3D object detection. It exploits high-level common sense properties from unlabeled data and bootstraps a dynamic object detector via repeated self-training. Despite promising performance, it requires excessive training time, which makes it difficult for practical use and development.

**Reward Fine-Tuning for Model Alignment**. Recently, foundation models [6, 29, 11, 44, 35, 37] have been shown to achieve strong performance in diverse tasks [5, 50], but sometimes produce outputs that do not align with human values [18, 30, 10]. A line of research aims to improve model alignment under the paradigm of Reinforcement Learning with Human Feedback (RLHF). Some pioneering works [41, 31, 33] learn a reward model and train foundation models with Proximal Policy Optimization (PPO) [38], but PPO is often expensive and unstable to train, and more importantly, requires a probabilistic output on the action space. This makes it hard to use for the object detection setting, which primarily uses regression-based losses. Reward ranked finetuning [27, 13] is a simplified alternative paradigm. It samples from a foundation model itself, filters generations using the reward model, and conducts supervised finetuning with the filtered generations.

## 3 Discovery from Reward-Incentivized Finetuning

Our framework, DRIFT, is inspired by the recent success of reward ranked finetuning methods for improving model alignment in the NLP community [13, 27]. We show that a similar approach can be adapted for 3D object discovery.

**Problem Setup.** We wish to obtain a dynamic object detection model on LiDAR data, i.e., a model to detect mobile objects in the LiDAR point clouds, *without human annotations*. Let $P \in \mathcal{R}^{N \times 3}$ denote a $N$-point 3D point cloud captured by LiDAR from which we wish to discover objects. We assume inputs of *unlabeled* point clouds collected by a car equipped with synchronized sensors including LiDAR (for point clouds) and GPS/INS (for accurate position and orientation). Since no annotation is involved, such a dataset is easy to acquire from daily driving routines; we additionally assume it to cover some locations with *multiple* scans at different times for computation of PP-score.

**Dynamic Point Proposals.** DRIFT leverages prior works that use unsupervised point clouds to extract foreground-background segmentation proposals. While many works [22, 3] have promising dynamic foreground segmentation results, in this work we rely on point *Persistency Prior score* (PP-score) [55] for its accuracy and leave the extension of other proposal methods to future work. For the purpose of this research, dynamic foreground points constitute LiDAR points reflecting off traffic participants (e.g. cars, bicyclists, pedestrians).

Using historical LiDAR sweeps collected at nearby locations of our point cloud $P$, the PP-score [3, 55] $\tau(P) \in [0, 1]^N$ can provide an informative estimate on the per-point persistence, i.e., whether a point belongs to persistent background or not. The PP-score is defined as the normalized entropy over past point densities, based on the assumption that background space such as ground, trees, and buildings tend to exhibit consistent point densities across different LiDAR scans (high entropy), whereas points associated with mobile objects exhibit high density only if an object is present (low entropy).

### 3.1 Rewarding "Good" Dynamic Boxes

We first establish a reward function that evaluates the quality of a set of bounding boxes for dynamic objects in a scene. We denote a set of $M$ dynamic objects bounding boxes as $\mathcal{B} = \{b_1, \ldots, b_M\}$, where each bounding box $b_i$ is represented as an upright box with parameters $(x_i, y_i, z_i, w_i, l_i, h_i, \theta_i)$, defining the box's center, width, length, height, and z-rotation, respectively. The scoring function $R$ scores the validity of the bounding boxes, given the observed point cloud $P$. In practice, a reward

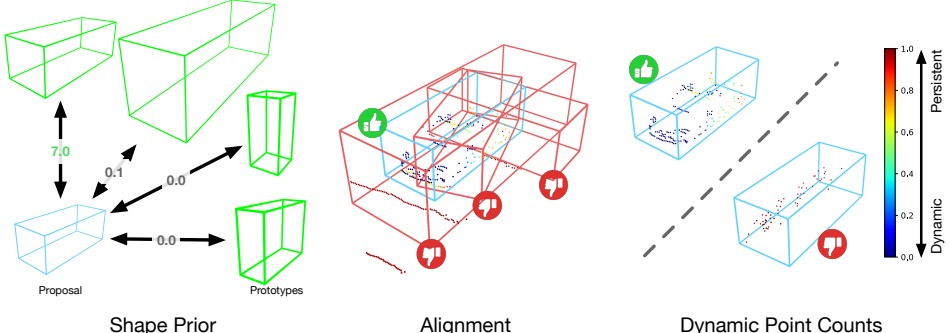

Shape Prior       Alignment       Dynamic Point Counts

Figure 2: **Illustration of the reward components.** The reward encourages boxes that have proper shape and alignment, and capture more dynamic points and few background points.

function that is positively correlated with IoU should suffice. We present our proposed reward function which aims to capture only dynamic points, filter nonsensical boxes, enforce correct size, and encourage proper box alignment to the captured dynamic points.

**Shape Prior Reward.** We enforce a box to not deviate significantly from a set of prototype sizes $\mathcal{S} = \{(\overline{w}_1, \overline{l}_1, \overline{h}_1), \ldots, (\overline{w}_C, \overline{l}_C, \overline{h}_C)\}$ (Fig. 2 left). We assume the shape prior distribution is a mixture of $C$ isotropic Gaussians with mixture weights $\pi_i$, diagonal variances $\Sigma_i$, and corresponding means as $(\overline{w}_i, \overline{l}_i, \overline{h}_i)$. These low-level statistics may be acquired directly from the dataset, or from vehicle specs and sales data available online [47]. In practice, we scale the mixtures such that the probabilities at the Gaussian means are equal for stability reasons. With this, the shape prior reward for box $\boldsymbol{b}$ is computed as:

$$R_{\text{shape}}(\boldsymbol{b}) = P_{\mathcal{S}}(\boldsymbol{b}). \tag{1}$$

**Alignment Reward.** Due to the nature of LiDAR sensing, the points will mostly fall on the lateral surfaces of an object. Therefore, a well-formed box should have dynamic points approximately close to the *boundary* of a box (Fig. 2 middle). As [55] shows, PP-score allows for easy separation of dynamic and persistent background points. Let $\boldsymbol{P}_{\text{dyn}}$ denote the set of dynamic points, and let $\boldsymbol{P}_{\text{bg}}$ be that of background points. In practice, since the PP-score is an approximation of ground-truth persistence, we define $\boldsymbol{P}_{\text{dyn}} = \{\boldsymbol{p}|\tau(\boldsymbol{p}) < 0.6\}$ and $\boldsymbol{P}_{\text{bg}} = \{\boldsymbol{p}|\tau(\boldsymbol{p}) \geq 0.9\}$.

Given a box $\boldsymbol{b}$, we denote all points within and close to the box as $\mathcal{O}(\boldsymbol{b})$. Only points within $\mathcal{O}(\boldsymbol{b})$ contribute to the reward of $\boldsymbol{b}$. In practice, we let $\mathcal{O}(\boldsymbol{b})$ consist of all points within a $\times 2$ scaled up version of $\boldsymbol{b}$ (with identical center and rotation). To score a box $\boldsymbol{b}$, we design a reward function that identifies how "typical" the dynamic points within $\mathcal{O}(\boldsymbol{b})$ are. For each dynamic point $\boldsymbol{p} \in \mathcal{O}(\boldsymbol{b}) \cap \boldsymbol{P}_{\text{dyn}}$, we compute the scaling factor $s_{\boldsymbol{p},\boldsymbol{b}}$ required so that the rescaled box touches $\boldsymbol{p}$ with one of its sides; i.e., if a point is inside the box, the box would have to be scaled down ($s_{\boldsymbol{p},\boldsymbol{b}} < 1$) to touch the point, if the point is outside it must be scaled up ($s_{\boldsymbol{p},\boldsymbol{b}} > 1$). We assume that $s_{\boldsymbol{p},\boldsymbol{b}}$ roughly follows a Gaussian distribution centered near the box boundary, and visualize the actual distribution in Fig. 3.

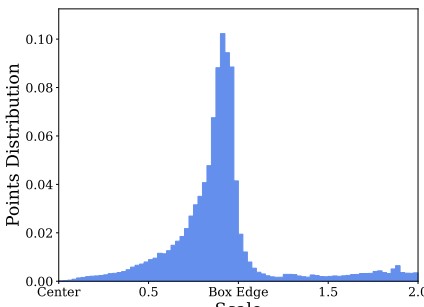

Figure 3: **Distribution of dynamic points near ground truth bounding boxes.** We observe that dynamic points near bounding boxes fall in an approximate Gaussian distribution centered near box edges ($s_{\boldsymbol{p},\boldsymbol{b}} \approx 0.8$).

We define our reward as the likelihood under this Gaussian distribution over scaling parameters. We approximate it as a Gaussian with hyper-parameters mean $\mu_{\text{scale}}$ and a variance $\sigma_{\text{scale}}$. Our reward is the product of the probability of each point in $\mathcal{O}(\boldsymbol{b})$:

$$R_{\text{align}}(\boldsymbol{b}) = \prod_{\boldsymbol{p} \in o(\boldsymbol{b}) \cap \boldsymbol{P}_{\text{dyn}}} \mathcal{N}(s_{\boldsymbol{p},\boldsymbol{b}}|\mu_{\text{scale}}, \sigma_{\text{scale}}). \tag{2}$$

**Common Sense Heuristics and Filtering.** Lastly, a proper bounding box must capture the dynamic points, and avoid capturing the background points (Fig. 2, right). This heuristic can be encoded by a

simple weighted point count for each bounding box $\boldsymbol{b}$:

$$R_{\text{count}}(\boldsymbol{b}) = \lambda_{\text{dyn}} \cdot |\boldsymbol{P}_{\text{dyn}} \cap \mathcal{O}(\boldsymbol{b})| - \lambda_{\text{bg}} \cdot |\boldsymbol{P}_{\text{bg}} \cap \mathcal{O}(\boldsymbol{b})|.$$

Intuitively, it assigns a reward in proportion to the number of dynamic points captured by the box, and a penalty in proportion to the number of background points captured.

Furthermore, boxes violating common sense should be assigned a low reward. We filter boxes that are too high up or too low from the ground, including those with too few dynamic points, or are too small or too large by directly assigning a reward of 0. In practice, we filter boxes that contain fewer than 4 dynamic points, or that have more than 80% persistent points, similar to [55].

In summary, the reward function is designed to be

$$R(\boldsymbol{b}) = \begin{cases} \lambda_{\text{shape}} \cdot R_{\text{shape}}(\boldsymbol{b}) + \lambda_{\text{align}} \cdot R_{\text{align}}(\boldsymbol{b}) + R_{\text{count}}(\boldsymbol{b}) & \text{if obeys common sense,} \\ 0 & \text{otherwise.} \end{cases} \tag{3}$$

## 3.2 Exploration Strategy for Improved Discovery

We assume a simple exploration strategy for identifying good box proposals. Given a set of current object proposals given by the detector, we locally perturb the boxes in the output space:

$$\mathcal{B}_{\text{explore}} \sim \mathcal{P}(\mathcal{B}_0, \sigma), \tag{4}$$

where $\mathcal{B}_{\text{explore}}$ is the set of explored boxes, perturbed from the model proposals $\mathcal{B}_0$. For each box $\boldsymbol{b} \in \mathcal{B}_0$, a set of explored boxes are sampled according to a standard Gaussian noise along the position and size dimensions and uniform noise for orientation:

$$\boldsymbol{b}_{\text{explore}}^{\text{center}} \sim \mathcal{N}(\boldsymbol{b}_0^{\text{center}}, \sigma^2 \boldsymbol{I}), \ \boldsymbol{b}_{\text{explore}}^{\text{size}} \sim \mathcal{N}(\boldsymbol{b}_0^{\text{size}}, \sigma^2 \boldsymbol{I}), \ \boldsymbol{b}_{\text{explore}}^{\theta} \sim \mathcal{U}(\boldsymbol{b}_0^{\theta} - \sigma, \boldsymbol{b}_0^{\theta} + \sigma). \tag{5}$$

Furthermore, to encourage proposals of boxes in foreground regions, we take inspiration from [56, 40] and re-use PP-score as *point-level* semantic segmentation (foreground vs background) labels, with which the detector is encouraged to propose boxes at points that have low PP-scores (i.e. are likely to be foreground points). Following [56], for each point $\boldsymbol{p}_i \in \boldsymbol{P}$ with prediction $\hat{\boldsymbol{y}}_i$, we assign its target classification label $\boldsymbol{y}_i$ as:

$$\boldsymbol{y}_i = \begin{cases} 1 & \text{if } \tau(\boldsymbol{p}_i) < \tau_L \text{ or } \hat{\boldsymbol{y}}_i = 1, \\ 0 & \text{otherwise.} \end{cases} \tag{6}$$

---

**Algorithm 1** Reward-Incentivized Finetuning

**Input:** Base object detector $f_\theta$, *unlabeled* LiDAR dataset $\mathcal{D}$, exploration noise $\sigma$, sample size $n$, filter budget $k$, reward function $R$, PP-scores $\tau$.
**repeat**
    $\boldsymbol{P} \sim \mathcal{D}$            ▷ Sample point cloud.
    $\mathcal{B}_0 \leftarrow f_\theta(\boldsymbol{P})$   ▷ Run detector for proposals.
    $\mathcal{B}_{\text{explore}} \sim \mathcal{P}(\mathcal{B}_0, \sigma)$     ▷ Sample $n$ boxes.
    $\boldsymbol{r} \leftarrow \{R(\boldsymbol{b})\}_{\boldsymbol{b} \in \mathcal{B}_{\text{explore}} \cup \mathcal{B}_0}$   ▷ Score boxes in set.
    $\mathcal{B} \leftarrow \text{NMS}(\mathcal{B}_{\text{explore}} \cup \mathcal{B}_0, \boldsymbol{r})$
    $\mathcal{B}_{\text{top}} \leftarrow \text{Filter}(\mathcal{B}, \boldsymbol{r}, k)$   ▷ Keep top $k$% boxes
    Update $\theta$ with $\mathcal{B}_{\text{top}}$ for 1 step
**until** converged

---

In effect, this encourages all non-persistent points (i.e., low $\tau(\boldsymbol{p}_i)$) to propose boxes near dynamic regions for better exploration.

## 3.3 Reward-Incentivized Finetuning

The reward function $R$ allows us to quickly evaluate proposed bounding boxes $\mathcal{B}$ and the task of 3D object discovery could be reduced to an optimization problem on the total reward in box set space:

$$\mathcal{B}^* = \arg\max_{\mathcal{B}} \sum_{\boldsymbol{b} \in \mathcal{B}} R(\boldsymbol{b}), \tag{7}$$

where the sum is taken over the boxes in the set $\mathcal{B}$. Although a direct optimization for $\mathcal{B}^*$ is not plausible due to the non-polynomial search space and discontinuity in $R$, $R$ can serve as effective guidance to facilitate model finetuning. The underlying intuition is similar to curriculum learning [4, 28, 46]: the object detection model takes small steps to improve from its current predictions towards $\mathcal{B}^*$ by following the direction provided by the maximum $R$ direction in a local space.

As illustrated in Alg. 1, in each training iteration, we first let the object detector perform inference on a point cloud $\boldsymbol{P}$ and propose a set of dynamic objects $\mathcal{B}_0$ in the scene. To explore directions

of improvement with the non-differentiable reward function $R$, we sample $n$ boxes from $\mathcal{B}_0$ (with replacement) and add an *i.i.d.* Gaussian noise on their location and size, and an uniform noise on orientation following Eq. 4. These sampled boxes are then ranked by the reward function $R$, in which the top $k$ non-overlapping boxes are selected by Non-Maximum Suppression (NMS) as training targets to finetune the object detector. Note that since DRIFT treats the model training/inference procedures as black boxes, it can be applied to any 3D object detection model.

In practice, it is observed that neural networks can acquire task knowledge from imperfect demonstrations [16, 51, 32]. MODEST [55] pre-trained the 3D object detector on noisy seed labels produced by DBSCAN [15] clustering on spatial and PP-score. We follow [55] and initialize our 3D object detector a model trained with discovered seed labels.

# 4 Experiments

**Datasets.** We experimented with two different datasets: Lyft Level 5 Perception dataset [24] and Ithaca-365 dataset [12]. To the best of our knowledge, these are the two publicly available datasets that contain multiple traversals of multiple locations with accurate 6-DoF localization and 3D bounding box labels for traffic participants.

In the Lyft dataset, we experiment with the same split provided by [55], where the train set and test set are geographically separated. It consists of 11,873 train scenes and 4,901 test scenes. For the Ithaca365 dataset, we experimented with the full dataset which consists of 57,107 scenes for training and 1,644 for testing. For both datasets, we do not use any human-annotated labels in training. To show the generalizability of our method, we conduct the development on the Lyft dataset, i.e., all the hyperparameters of our approach are finalized through experiments on Lyft, and we use the exact same set of hyperparameters for all experiments in Ithaca365.

**Evaluation.** Following [55], we combine all traffic participants to a single mobile object class and evaluate the detector's performance on this class. Note that the labels are not used during training but solely for evaluation. For Lyft, we report the mean average precision (mAP) of the detector with the intersection over union (IoU) thresholds at 0.5 and 0.7 in bird-eye-view perspective. Note that mAP at 0.7 IoU threshold is a stricter and harder metric and was not evaluated in [55], and we include it to emphasize the effectiveness of our method. For Ithaca365, we adopt metrics similar to those in [7]: we evaluate mean average precision (mAP) for dynamic objects under $\{0.5, 1, 2, 4\}$m thresholds that determine the match between detection and ground truth; we also compute 3 types of true positive metrics (TP metrics), including ATE, ASE and AOE for measuring translation, scale and orientation errors. These TP metrics are computed under a match distance threshold of 2m; additionally, we also compute a distance-based breakdown (0-30m, 30-50m, 50-80m) for these metrics.

**Implementation.** We use PointRCNN [40] as our default architecture and we use the implementation provided by OpenPCDet [42]. We train DRIFT with 120 epochs in Lyft and 30 epochs in Ithaca365 as the default setting, and observe that the performance generally improves with more training epochs (Fig. 1). We use $\lambda_{shape} = 1$, $\lambda_{align} = 1$, $\lambda_{dyn} = 0.001$ and $\lambda_{bg} = 0.001$. We use $\mu_{scale} = 0.8$ and $\sigma_{scale} = 0.2$ for the alignment reward. We define the shape priors based on four typical types of traffic participants: Car, Pedestrian, Cyclist, and Truck. Specifically, we use the mean and standard deviation of box sizes of each class in the Lyft dataset, but we show that they generalize well to other domains like Ithaca365 and are not sensitive (Tab. 2) The exact prototype sizes $\mathcal{S}$ and other implementation details can be found in the supplementary materials.

**Baselines.** To the best of our knowledge, MODEST [55] is the only prior work on this problem and we compare our method DRIFT against it with various variants of MODEST: (1) No Finetuning: the model trained with seed labels from PP-score without repeated self-training (MODEST (R0)) in [55]; (2) Self-Training ($i$ ep): the model initialized with (1) and self-trained with $i$ epochs without PP-score filtering; (3) MODEST ($i$ ep): the model initialized with (1) and self-trained with $i$ epochs with PP-score filtering (full MODEST model). For self-training in (2) and (3), we adopt 60 epochs for each self-training round in the Lyft dataset (same as that in [55]) and 30 epochs for the Ithaca365 dataset. To ensure a fair comparison, DRIFT is also initialized from (1) and use the same detector configurations as the baselines. Following [55], we also compare with the supervised counterparts trained with human-annotated labels from the same dataset (Lyft or Ithaca365) and from another out-of-domain dataset (KITTI).

Table 1: **Detection performance on Lyft.** DRIFT outperforms both baselines with 10% training time, and approaches the performance of the out-of-domain supervised detector trained on KITTI. Please refer to the setup of Sec. 4 for the metrics.

| Method | mAP IoU @ 0.5 (↑) | | | | mAP IoU @ 0.7 (↑) | | | |
|---|---|---|---|---|---|---|---|---|
| | 0-30 | 30-50 | 50-80 | 0-80 | 0-30 | 30-50 | 50-80 | 0-80 |
| No Finetuning | 44.1 | 21.1 | 1.2 | 23.9 | 24.4 | 6.0 | 0.1 | 10.5 |
| Self-Train. (60 ep) | 50.0 | 29.0 | 3.4 | 28.6 | 32.5 | 10.0 | 0.3 | 14.0 |
| Self-Train. (600 ep) | 56.7 | 41.1 | 9.1 | 37.2 | 35.1 | 20.7 | 1.6 | 19.9 |
| MODEST (60 ep) | 49.6 | 29.7 | 3.4 | 28.8 | 31.3 | 10.2 | 0.3 | 14.4 |
| MODEST (600 ep) | 56.4 | **45.4** | 11.3 | 39.6 | 33.6 | 18.6 | 1.4 | 18.8 |
| DRIFT (30 ep) | 60.1 | 40.2 | 9.1 | 38.3 | 39.0 | 24.2 | 3.6 | 23.1 |
| DRIFT (60 ep) | 60.3 | 43.8 | 14.6 | 41.8 | 42.0 | 29.2 | 5.8 | 26.7 |
| DRIFT (120 ep) | **61.4** | 45.1 | **21.7** | **45.3** | **42.7** | **31.7** | **9.9** | **29.6** |
| Sup. on KITTI | 71.9 | 49.8 | 22.2 | 49.9 | 47.0 | 26.2 | 6.4 | 27.9 |
| Sup. on Lyft | 76.9 | 60.2 | 37.5 | 60.4 | 62.7 | 50.9 | 28.2 | 48.5 |

Table 2: **Detection performance on Ithaca365.** We observe DRIFT outperforms both baselines with significantly less training time. Please refer to the setup of Sec. 4 for the metrics.

| Method | mAP (↑) | | | | Errors 0-80m (↓) | | |
|---|---|---|---|---|---|---|---|
| | 0-30 | 30-50 | 50-80 | 0-80 | ATE | ASE | AOE |
| No Finetuning | 18.7 | 4.8 | 0.0 | 7.7 | 1.17 | 0.60 | 1.64 |
| Self-Train. (30 ep) | 25.9 | 9.2 | 1.2 | 12.4 | 1.08 | 0.62 | 1.57 |
| Self-Train. (300 ep) | 16.3 | 3.6 | 1.8 | 6.8 | 1.19 | 0.74 | 1.57 |
| MODEST (30 ep) | 14.6 | 0.7 | 0.0 | 3.7 | 0.83 | 0.52 | 1.53 |
| MODEST (300 ep) | 27.5 | 26.3 | 21.0 | 27.1 | 1.06 | 0.67 | **1.09** |
| DRIFT (15 ep) | 39.1 | 24.3 | 17.7 | 28.0 | 0.73 | **0.33** | 1.23 |
| DRIFT (30 ep) | **47.1** | **31.2** | **22.9** | **35.1** | **0.49** | 0.35 | 1.20 |
| Sup. on KITTI | 59.8 | 28.3 | 4.0 | 32.0 | 0.26 | 0.22 | 0.46 |
| Sup. on Ithaca365 | 75.7 | 48.3 | 22.6 | 51.5 | 0.18 | 0.13 | 0.33 |

**Dynamic Object Detection Results.** We report the performance of DRIFT and baseline detectors on Lyft in Tab. 1, and show the performance over the training epochs in Fig. 1. We report the performance on Ithaca365 in Tab. 2. Notably, DRIFT demonstrates significantly faster learning and strong performance. It provides more than $10\times$ speedup as compared to the baselines. On Lyft, DRIFT's performance at 60 epochs already surpasses the performance of both baselines at 600 epochs (10 self-training rounds) and approaches the performance of the out-of-domain supervised detector trained on KITTI [19]. On Ithaca365, its performance at 30 epochs significantly surpasses both baselines trained at 300 epochs. It even outperforms the out-of-domain supervised detector trained on KITTI in mAP.

Table 3: Analysis on rewards of boxes produced by different detectors. We report mean and std of box reward on the Lyft dataset.

| Models | Mean | StD. |
|---|---|---|
| Rand. Boxes | 0.02 | 0.15 |
| Self-Train. | 0.44 | 0.57 |
| MODEST | 0.57 | 0.58 |
| DRIFT | 0.61 | 0.64 |
| Ground Truth | 1.00 | 0.42 |

Observe that the self-training performance starts collapsing with more rounds of self-training, and does not continue to improve.

Fig. 4 visualizes the detection on two scenes. Ground truth boxes are colored in green, predictions from the detector without fine-tuning are in yellow, and predictions from DRIFT are in red. We observe that the detector without fine-tuning occasionally produces false positive predictions, produces boxes with incorrect sizes, or misses moving objects, while DRIFT produces accurate detection.

**Rewards ablations.** We report the average reward per box for ground truth boxes, random boxes, and predicted boxes from different detectors in Tab. 3. The ground truth boxes have the highest rewards on average, while the random boxes have the lowest. This indicates that the reward reasonably reflects

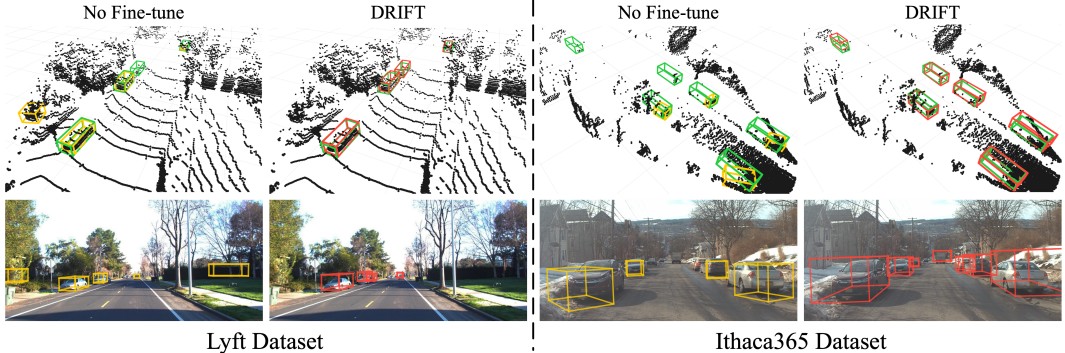

| No Fine-tune | DRIFT | | No Fine-tune | DRIFT |
| Lyft Dataset | | | Ithaca365 Dataset | |

Figure 4: **Visualization of detections.** Qualitative results on two scenes from Lyft [24] and Ithaca365 [12] datasets. Ground truth boxes are labeled with green in the LiDAR figures and predictions without fine-tuning and DRIFT are in yellow and red, respectively. We observe DRIFT learns to produce accurate detection with correct shape and localization.

Table 4: Ablation on the reward components. We report the mAP (0-80m) on Lyft. Removing components significantly degrades performance.

| Filter | Shape | Align. | mAP (0 - 80m) | |
| | | | IoU 0.5 | IoU 0.7 |
| --- | --- | --- | --- | --- |
| ✓ | | | 0.6 | 0.0 |
| ✓ | ✓ | | 0.0 | 0.0 |
| | ✓ | ✓ | 0.0 | 0.0 |
| ✓ | | ✓ | 22.4 | 3.2 |
| ✓ | ✓ | ✓ | 38.3 | 23.1 |

Table 5: Ablation on the alignment reward's $\mu_{scale}$. We report the mAP (0-80m) on the Lyft dataset. We show the detection performance is not sensitive to the choice of $\mu_{scale}$.

| $\mu_{scale}$ | mAP (0 - 80m) | |
| | IoU 0.5 | IoU 0.7 |
| --- | --- | --- |
| 0.8 | 38.3 | 23.1 |
| 0.9 | 38.1 | 23.4 |

Table 6: Ablation on the alignment reward's $\sigma_{scale}$. We report the mAP (0-80m) on the Lyft dataset. The detection performance is not sensitive to the variance.

| $\sigma_{scale}$ | mAP (0 - 80m) | |
| | IoU 0.5 | IoU 0.7 |
| --- | --- | --- |
| 0.1 | 34.2 | 20.9 |
| 0.2 | 38.3 | 23.1 |
| 0.3 | 38.1 | 20.0 |

the quality of the bounding box. And we observe the boxes predicted by DRIFT have higher rewards than those predicted by the baseline detectors.

Ablation study on the components of our reward is presented in Tab. 4, and visualization is shown in Fig. 5. Detection performance significantly drops when we remove one or more of the components. For example, when only common sense filtering is used, the detector just predicts boxes around foreground points. Without the shape prior reward, the detector predicts boxes with incorrect sizes.

Ablations Tab. 5 and Tab. 6 present the sensitivity analysis of the choices of $\mu_{scale}$ and $\sigma_{scale}$. DRIFT achieves stable performance across reasonable choices of $\mu_{scale}$ and $\sigma_{scale}$, showing the robustness of our method.

**Exploration.** We study the necessity of the exploration component and the effect of incorporating other sources for box sampling. In Tab. 7, we compare no exploration to: (1) sampling 200 boxes from box predictions, (2) sampling 100 from proposals near dynamic points and 100 from predictions, and (3) sampling 100 from seed labels and 100 from predictions. Observe that the exploration component is crucial for our method; by performing local exploration instead of simply updating from its own predictions, DRIFT avoids confirmation bias and ensures that labels improve over what it predicts. Furthermore, results show that sampling from the box predictions is sufficient for obtaining good performance; other sources do not provide obvious benefits.

We also explore the effects of modifying the exploration strategy. Tab. 8 compares the detector performance of using sample size of 50, 100 and 200, and noise scale $\sigma$ (i.e.variation) of 0.3 vs. 0.6. Each detector is trained for 30 epochs. At noise scale 0.3, increasing the sample size from 50 to 200 significantly improves the detection performance. Using noise scale 0.6 significantly reduces the detection performance, indicating that smaller noise may be preferable.

**Filtering Budget of the Ranked Boxes.** We study the effect of the choice of top $k\%$ for filtering boxes by reward ranking. Tab. 9 presents the detection performance with top 55%, 65%, 75% and 85%. DRIFT is robust to the choice of $k$, with slightly decreased performance when $k$ is too high.

Table 7: Detection performance with additional sampling sources in exploration. Sampling near the predictions is sufficient; additional sources do not provide obvious benefits. Trained for 60 ep.

| | mAP (0 - 80m) | |
| --- | --- | --- |
| | IoU 0.5 | IoU 0.7 |
| No Exploration | 0.0 | 0.0 |
| Sample near pred. | 41.8 | 26.7 |
| + near dynamic | 39.3 | 22.6 |
| + init. seed labels | 39.5 | 22.8 |

Table 8: Ablation on exploration parameters. Large sampling size and moderate noise scale are preferable.

| Noise | Samples | mAP (0 - 80m) | |
| --- | --- | --- | --- |
| | | IoU 0.5 | IoU 0.7 |
| 0.3 | 50 | 32.9 | 14.3 |
| | 100 | 36.4 | 18.9 |
| | 200 | 38.3 | 23.1 |
| 0.6 | 50 | 5.6 | 0.0 |
| | 100 | 17.2 | 1.3 |
| | 200 | 0.6 | 0.0 |

Table 9: Ablation on the choice of top $k\%$ for box filtering by reward ranking. DRIFT is robust to different values of $k$.

| Top K | mAP (0 - 80m) | |
| --- | --- | --- |
| | IoU 0.5 | IoU 0.7 |
| 55% | 37.0 | 21.0 |
| 65% | 38.9 | 23.0 |
| 75% | 38.3 | 23.1 |
| 85% | 35.8 | 20.2 |

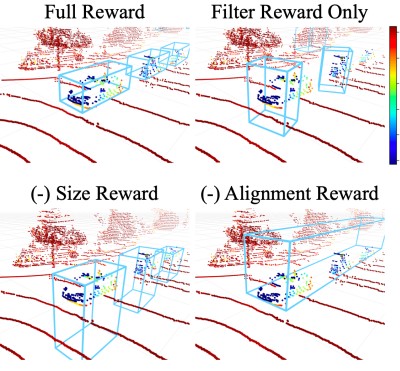

Figure 5: **Visualization of reward ablation.** Removing components leads to predictions with incorrect shape or position.

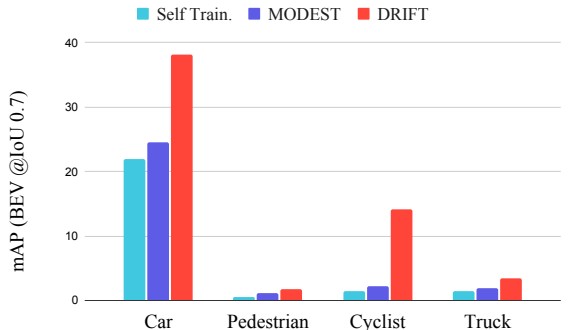

Figure 6: **Per-class BEV mAP at IoU 0.7.** We assign each predicted box to the class with the most similar prototype size. DRIFT outperforms the baselines for all classes.

**Extension to Detection with Classes.** Our prototype sizes are defined by different classes of traffic participants. Thus, given a predicted box from a class agnostic detector, we can compute the likelihood of its size under the Gaussian prior of each prototype size, and assign it to the class with the highest likelihood. Fig. 6 presents the per-class performance of DRIFT and baselines under such assignment. All detectors are trained for 60 epochs. DRIFT outperforms both baselines, with especially significant improvement for the Car and Cyclist classes. More details can be found in the supplementary materials.

## 5  Discussion and Conclusion

In this work, we propose a framework, DRIFT, to tackle object discovery without labels. Instead of requiring expensive 3D bounding box labels, our method utilizes succinctly described heuristics as a reward signal to initialize and subsequently fine-tune an object detector. To optimize this non-differentiable reward signal, we propose a simple but very effective reward finetuning framework, inspired by recent successes of reinforcement learning in the NLP community. Compared to prior self-training based methods [55], such a framework is an order of magnitude faster to train, while achieving higher accuracy. Traditional self-training iteratively generates pseudo-labels and retrains the model, requiring convergence before generating the next set of pseudo-labels. In general, training a detector to mimic pseudo-labels can lead to undesirable artifacts, further amplified by repeated training (confirmation bias). DRIFT addresses this issue by leveraging reinforcement learning principles, where the exploration component is crucial. Our method avoids confirmation bias by performing local exploration and ensures that labels improve over what it predicts. Thus, DRIFT is able to perform updates per-training iteration as opposed to per self-training round, which allows it to converge significantly faster and achieve higher performance.

**Limitations and Future Works.** One limitation is that the current framework is geared explicitly towards dynamic objects. Static objects would require different heuristics, not based on PP-scores. Similarly, currently we restricted our framework entirely to LiDAR signals. However, the reward

based framework is extremely flexible, and could easily be extended to other data modalities. For example, one could use image features to help identify objects inside of box proposals. Although in supervised settings image features have typically not added much in to the higher resolution LiDAR point clouds, in our unsupervised setting it is certainly possible that pixel information can help disambiguate objects from background. Further, we plan to explore the use of reward fine-tuning for other vision applications beyond object discovery.

## Acknowledgments and Disclosure of Funding

This research is supported in part by grants from the National Science Foundation (III-2107161, IIS 2144117, and IIS-1724282), Nvidia Research, and DARPA Geometries of Learning (HR00112290078). We also thank Wei-Lun Chao, Yihong Sun, Travis Zhang, and Junan Chen for their valuable discussion.

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

# Supplementary Material:
# Reward Finetuning for Faster and More Accurate Unsupervised Object Discovery

## S1 Additional Implementation Details

We train DRIFT on four NVIDIA RTX A6000 GPUs, with batch size 10 per GPU. The values we use for the shape priors are reported in Tab. S10. In practice, we scale the mixtures such that the probabilities at the Gaussian means are equal for stability reasons. The values in Tab. S10 are computed from the box shapes in the Lyft dataset, but the method generalize well to other domains like Ithaca365. Bear in mind, all models trained on Ithaca365 directly uses the hyperparameters found in Lyft, suggesting the generalizability of our method. DRIFT achieves stable performance when alternative values are used (Tab. S16).

Table S10: Shape prior values used in our implementation.

| Mixture Component | Width $w$ | | Length $l$ | | Height $h$ | |
|---|---|---|---|---|---|---|
| | Mean | StD. | Mean | StD. | Mean | StD. |
| 1 (Car) | 1.911 | 0.162 | 4.745 | 0.559 | 1.711 | 0.248 |
| 2 (Pedestrian) | 0.780 | 0.153 | 0.797 | 0.182 | 1.745 | 0.177 |
| 3 (Truck) | 2.832 | 0.278 | 9.403 | 3.145 | 3.299 | 0.430 |
| 4 (Cyclist) | 0.613 | 0.256 | 1.752 | 0.326 | 1.364 | 0.343 |

## S2 Extended Quantitative Results

### S2.1 Full Tables from Main Paper

We provide results on all ranges for all ablation tables shown in the the main paper. In Tab. S11, we report the results of the alignment reward ablation on $\mu_{scale}$. In Tab. S12, we report ablation results on reward components. In Tab. S13 and Tab. S14 we report ablation on exploration method and sample counts, respectively. In Tab. S15, we report ablation results for varying the filter budget.

Table S11: Ablation on the alignment reward's $\mu_{scale}$. We report the mAP (0-80m) on the Lyft dataset. This corresponds with Table 5 of the main paper.

| Alignment Reward | mAP IoU 0.5 | | | | mAP IoU 0.7 | | | |
|---|---|---|---|---|---|---|---|---|
| | 0-30 | 30-50 | 50-80 | 0-80 | 0-30 | 30-50 | 50-80 | 0-80 |
| $\mu = 0.8$ | 60.1 | 40.2 | 9.1 | 38.3 | 39.0 | 24.2 | 3.6 | 23.1 |
| $\mu = 0.9$ | 58.3 | 39.2 | 10.1 | 38.1 | 37.1 | 26.0 | 4.4 | 23.4 |

### S2.2 Ablation on Shape Prior Reward

We additionally include an ablation study on the shape-prior reward in Tab. S16. We show that it is not sensitive to the choice in variance used for each class. We assume that the means of the shape priors are user defined.

### S2.3 Lower IoU and Recall Evaluation

We report mAP results at IoU 0.25 match in Tab. S17. In particular, IoU 0.25 metric evaluates "localization", i.e. if there is a bounding box with a very small overlap with the ground truth. However,

Table S12: Ablation on the reward components. We report the mAP (0-80m) on the Lyft dataset. This table corresponds to Table 4 in the main paper.

| Filtering | Size Prior | Align. | mAP IoU 0.5 | | | | mAP IoU 0.7 | | | |
|---|---|---|---|---|---|---|---|---|---|---|
| | | | 0-30 | 30-50 | 50-80 | 0-80 | 0-30 | 30-50 | 50-80 | 0-80 |
| ✓ | | | 1.3 | 0.8 | 0.0 | 0.6 | 0.0 | 0.0 | 0.0 | 0.0 |
| ✓ | ✓ | | 0.0 | 0.0 | 0.0 | 0.0 | 0.0 | 0.0 | 0.0 | 0.0 |
| | ✓ | ✓ | 0.0 | 0.0 | 0.0 | 0.0 | 0.0 | 0.0 | 0.0 | 0.0 |
| ✓ | | ✓ | 41.2 | 18.2 | 1.7 | 22.4 | 5.3 | 5.0 | 0.3 | 3.2 |
| ✓ | ✓ | ✓ | 60.1 | 40.2 | 9.1 | 38.3 | 39.0 | 24.2 | 3.6 | 23.1 |

Table S13: Detection performance with additional sampling sources in exploration. Sampling near the predictions is sufficient; additional sources do not provide obvious benefits.

| | mAP IoU 0.5 | | | | mAP IoU 0.7 | | | |
|---|---|---|---|---|---|---|---|---|
| | 0-30 | 30-50 | 50-80 | 0-80 | 0-30 | 30-50 | 50-80 | 0-80 |
| Sample near pred. | 60.3 | 43.8 | 14.6 | 41.8 | 42.0 | 29.2 | 5.8 | 26.7 |
| + near dynamic | 59.8 | 40.9 | 10.6 | 39.3 | 39.1 | 22.8 | 3.2 | 22.6 |
| + init. seed labels | 60.3 | 40.7 | 10.4 | 39.5 | 38.4 | 23.9 | 3.3 | 22.8 |

our method excels at predicting the size and proper orientation, which is better captured at higher IoUs metrics. To summarize, as compared to the prior work, MODEST can localize boxes well, but it's not able to figure out the size; in contrast, our method does both well. As stated in the main paper, we use 0.5 and 0.7 to (1) following the KITTI and Lyft standards of reporting, and (2) emphasize the strength of our method.

In addition, we report recall metrics in Tab. S18. We see that the recall improves over additional training, which suggests that part of the improvement comes from being able to better detect bounding boxes by locating them in the scene.

## S2.4 Ablation on class mixtures

We include additional experiments on the class type mixtures that we use in the reward function of DRIFT to evaluate the robustness of our method. We report the results of using 4 (the ground truth number), 5, and 6 factors in the Gaussian mixture in Tab. S19.

# S3 Extended Qualitative Visualizations

We showcase additional qualitative results in Fig. S8. Observe that DRIFT improves significantly over the model without fine-tuning, and close to supervised performance, without any labels.

Table S18: **Recall by epoch.** We report class recall numbers at different training epochs. Observe that DRIFT training improves the recall.

| Epoch | Recall (0 - 80m) | |
|---|---|---|
| | IoU 0.5 | IoU 0.7 |
| 30 | 0.47 | 0.30 |
| 60 | 0.51 | 0.34 |
| 90 | 0.53 | 0.36 |
| 120 | 0.56 | 0.38 |

Table S14: Ablation on sampling hyperparameters for exploration. Large sampling size and moderate noise scale are preferable.

| Noise | Samples | mAP IoU 0.5 | | | | mAP IoU 0.7 | | | |
|---|---|---|---|---|---|---|---|---|---|
| | | 0-30 | 30-50 | 50-80 | 0-80 | 0-30 | 30-50 | 50-80 | 0-80 |
| | 50 | 57.3 | 31.3 | 4.8 | 32.9 | 26.0 | 14.4 | 1.4 | 14.3 |
| 0.3 | 100 | 59.8 | 36.2 | 7.0 | 36.4 | 33.0 | 18.9 | 1.8 | 18.9 |
| | 200 | 60.1 | 40.2 | 9.1 | 38.3 | 39.0 | 24.2 | 3.6 | 23.1 |
| | 50 | 17.8 | 1.2 | 0.0 | 5.6 | 0.1 | 0.0 | 0.0 | 0.0 |
| 0.6 | 100 | 39.5 | 10.4 | 0.2 | 17.2 | 4.0 | 0.3 | 0.0 | 1.3 |
| | 200 | 0.1 | 0.9 | 1.1 | 0.6 | 0.0 | 0.0 | 0.0 | 0.0 |

Table S15: Ablation on the choice of top $k\%$ for box filtering by reward ranking. DRIFT is robust to different values of $k$.

| Top $k\%$ | mAP IoU 0.5 | | | | mAP IoU 0.7 | | | |
|---|---|---|---|---|---|---|---|---|
| | 0-30 | 30-50 | 50-80 | 0-80 | 0-30 | 30-50 | 50-80 | 0-80 |
| 55% | 60.0 | 36.5 | 7.5 | 37.0 | 36.5 | 20.1 | 2.3 | 21.0 |
| 65% | 60.1 | 39.9 | 9.7 | 38.9 | 39.1 | 23.4 | 3.5 | 23.0 |
| 75% | 60.1 | 40.2 | 9.1 | 38.3 | 39.0 | 24.2 | 3.6 | 23.1 |
| 85% | 58.1 | 36.6 | 6.6 | 35.8 | 35.8 | 20.6 | 2.3 | 20.2 |

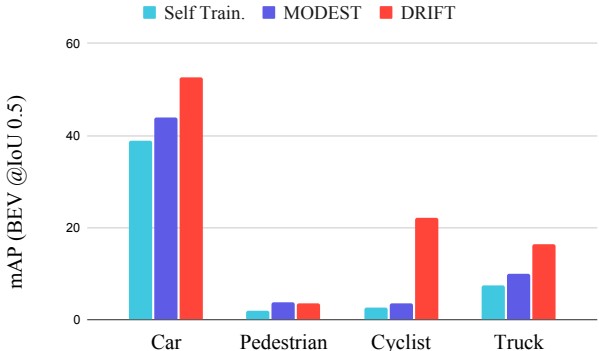

Figure S7: **Per-class BEV mAP at IoU 0.5.** We assign each predicted box to the class with the most similar prototype size. This corresponds to Figure 6 of the main paper.

Table S16: Ablation on variance of the class shape priors.

| Class Shape Var. | mAP IoU 0.5 | | | | mAP IoU 0.7 | | | |
|---|---|---|---|---|---|---|---|---|
| | 0-30 | 30-50 | 50-80 | 0-80 | 0-30 | 30-50 | 50-80 | 0-80 |
| $\sigma_i = 0.5$ StD. | 57.6 | 34.5 | 5.1 | 34.5 | 36.8 | 22.5 | 1.7 | 21.3 |
| $\sigma_i = 0.2$ StD. | 55.3 | 40.2 | 11.1 | 37.1 | 39.8 | 27.9 | 5.2 | 25.5 |
| True Variance | 60.1 | 40.2 | 9.1 | 38.3 | 39.0 | 24.2 | 3.6 | 23.1 |

Table S17: **Detection performance on the Lyft dataset.** We report the evaluation results at IoU 0.25. This metric is evaluating "localization", i.e. if there is a bounding box with a tiny overlap with the GT. Our method matches the baselines on this match threshold, and surpasses on the more strict IoU thresholds of 0.5 and 0.7.

|  | mAP IoU 0.25 (↑) | | | |
|---|---|---|---|---|
|  | 0-30 | 30-50 | 50-80 | 0-80 |
| No Finetuning | 63.5 | 34.9 | 6.0 | 37.5 |
| Self-Train. (60 ep) | 67.7 | 43.2 | 8.7 | 43.1 |
| Self-Train. (600 ep) | 67.7 | 48.0 | 13.3 | 45.5 |
| MODEST (60 ep) | 68.5 | 46.2 | 10.5 | 45.1 |
| MODEST (600 ep) | 73.6 | 56.8 | 21.0 | 53.6 |
| DRIFT (60ep) | 72.3 | 51.5 | 19.2 | 50.7 |
| DRIFT (120 ep) | 72.5 | 51.7 | 25.8 | 52.9 |
| Supervised on KITTI | 78.6 | 53.9 | 26.1 | 55.3 |
| Supervised on Lyft | 81.8 | 63.6 | 40.0 | 64.2 |

Table S19: **Ablation on the number of factors in the class mixture.** We test out additional class sizes other than the ground truth number of classes in the label set (car, pedestrian, cyclist, truck). Observe that the number of factors does not significantly affect the performance of DRIFT.

|  | mAP IoU 0.5 | | | | mAP IoU 0.7 | | | |
|---|---|---|---|---|---|---|---|---|
| Num. Factors | 0-30 | 30-50 | 50-80 | 0-80 | 0-30 | 30-50 | 50-80 | 0-80 |
| 4 (GT) | 60.1 | 40.2 | 9.1 | 38.3 | 39.0 | 24.2 | 3.6 | 23.1 |
| 5 | 59.8 | 37.9 | 7.4 | 37.0 | 38.6 | 24.7 | 2.9 | 23.0 |
| 6 | 58.5 | 38.3 | 6.1 | 36.0 | 39.1 | 23.9 | 2.6 | 22.7 |

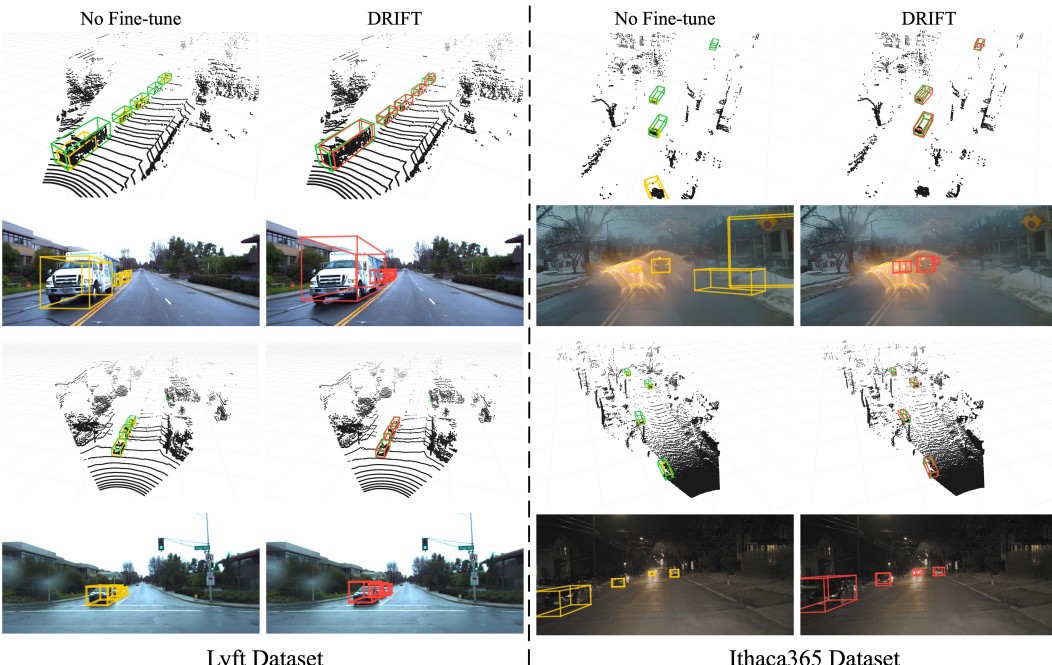

Figure S8: **Qualitative visualizations.** Additional visualizations of the DRIFT method, as compared to no fine-tuning.

