# OpenReview forum: "Reward Finetuning for Faster and More Accurate Unsupervised Object Discovery"
_NeurIPS.cc/2023/Conference — NeurIPS 2023 poster_

### Official Review · Reviewer_Baoq · 2023-06-12

**Soundness:** 3 good
**Presentation:** 3 good
**Contribution:** 3 good
**Rating:** 6
**Confidence:** 3

**Summary:**

The paper addresses the research question of unsupervised 3D object location detection from LIDAR data in autonomous driving scenes.

The proposed method, DRIFT, improves upon the MODEST baseline by incorporating heuristics for judging objectness likelihood and using them as rewards within a reinforcement learning framework.

DRIFT is evaluated on the LYFT and Ithaca benchmarks, demonstrating enhanced performance and training efficiency compared to the MODEST baseline.

**Before rebuttal**:

The paper presents a promising contribution and could be a valuable addition to NeurIPS this year. However, since my familiarity lies more in the domain of object detection from 2D scenes, I may not be able to provide detailed insights into the previous work in the LIDAR domain.

**Strengths:**

**S1.** The paper is well-written, well-executed, and effectively addresses the under-explored problem of unsupervised 3D object location detection in autonomous driving scenes. The proposed method, DRIFT, demonstrates the potential for real-life applications.

**S2.** The proposed idea is well-grounded and logical. The incorporation of studied priors based on shape, size, and location to judge objectness likelihood aligns with human intuition. The utilization of a feedback loop through reward-based fine-tuning to improve model performance over time is a sensible approach.

**S3.** The reported improvement achieved by the authors is significant. Despite training for only 30 epochs (compared to MODEST's 600 epochs), DRIFT outperforms the strong baseline on both datasets. Although there is still a noticeable gap when compared to supervised methods, the results demonstrate promising progress.

**Weaknesses:**

**W1.** It would have strengthened the paper to provide a context within the well-studied objectness literature [1-2]. The concept of objectness aims to identify low-level, generic cues that distinguish foreground regions from the background, such as edge distributions, boundary textures, and likely object size, shape, and location. The authors' goal aligns with this objective, albeit from a different modality (LIDAR). Drawing inspiration and techniques from the objectness literature could have been insightful.

**W2.** An explanation of how the authors generated the shape templates (priors, prototypes) would have been beneficial. It is unclear how these templates were derived or selected.

**W3.** Given the reliance on multiple heuristics in the method (which is reasonable and sound), it would have been valuable to investigate cross-dataset generalization, specifically from LYFT to Ithaca. Understanding how the measured statistics change across different domains and driving scenes would provide insights into the method's robustness and applicability.

[1] BING: Binarized normed gradients for objectness estimation at 300fps, https://mmcheng.net/mftp/Papers/ObjectnessBING.pdf

[2] Survey and Performance Analysis of Deep Learning Based Object Detection in Challenging Environments, https://www.mdpi.com/1424-8220/21/15/5116


**Questions:**

See the weaknesses. More like a suggestion:

Please state from the very start that your goal is to find generic (foreground vs. background) object regions, and NOT categorization. This maybe confusing to some readers like me, that when you use the term "Object Discovery", I look for a semanic categorization component as well.


**Limitations:**

Limitations are presented.

---

> ### Author Rebuttal · Authors · 2023-08-09
>
> Thank you for the positive feedback and appreciation of our contributions! We address individual questions below:
>
> > Include context within the well-studied objectness literature
>
> We thank the reviewer for pointing out the comprehensive works [1, 2] and the explanation of the concept of objectness. Indeed, objectness is aligned with our goals in this work, especially its goals to identify how likely something is an object in *any category*. In particular, we were surprised to learn about the metrics that are used in this field of work that’s explicitly designed to handle class-free proposal generation, although from a different modality. In the future, we will use inspiration from these works to evaluate the works we do in this direction. We thank the reviewer for bringing this subject to our attention and will include extensive discussion into objectness estimation as prior for our work.
>
> > Explanation of how shape templates were generated
>
> We show category-wise object size statistics (unit: meter) in the table below in the format of $\mu$ ($\sigma$):
> | Dataset | Category | Length | Height | Width |
> |--|:--:|:--:|:--:|:--:|
> | Lyft | Truck | 9.40 (3.15) | 3.30 (0.43) | 2.83 (0.27) |
> | Lyft | Cyclist | 1.75 (0.33) | 1.36 (0.34) | 0.61 (0.26) |
> | Lyft | Car | 4.74 (0.56) | 1.71 (0.25) | 1.91 (0.16) |
> | Lyft | Pedestrian | 0.80 (0.18) | 1.74 (0.18) | 0.78 (0.15) |
> | Ithaca365 | Truck | 6.09 (2.57) | 2.33 (0.77) | 2.22 (0.48) |
> | Ithaca365 | Cyclist | 1.75 (0.64) | 1.52 (0.51) | 0.71 (0.22) |
> | Ithaca365 | Car | 4.41 (0.22) | 1.55 (0.15) | 1.75 (0.12) |
> | Ithaca365 | Pedestrian | 0.60 (0.19) | 1.70 (0.14) | 0.61 (0.12) |
>
> We selected our templates of the shape priors as well as the number of prototypes based on the Lyft L5 dataset statistics, following prior self-driving works using the classes: cars, pedestrians, cyclists, and trucks (Supplementary, Tab. 1). In this work, we assume domain knowledge over object sizes for which we wish to discover. However, we extensively ablate how much the priors and prototypes affect the final results, and find that our method is not sensitive to the values (Supplementary, Tab. 8), and generalizes to the Ithaca365 dataset *without changing the values* (Table 2). In addition, we found the method is not sensitive to the number of prototypes. We include this ablation in the response to reviewer dARK and again below:
>
> | Num. Factors | IoU 0.5 | IoU 0.7 |
> |--------------|:-------:|:-------:|
> |       4 (GT) |   38.3  |   23.1  |
> |            5 |   37.0  |   23.0  |
> |            6 |   36.0  |   22.7  |
>
> Including additional prototypes at 10, 15-meters doesn’t significantly change the results, following the intuition that the method will search for the size prototype that best describes/fit the dynamic points.
>
> > Cross-dataset generalization of heuristics (Lyft to Ithaca365)
>
> We calculate the true $\mu_{scale}^*$ and $\sigma_{scale}^*$ from dataset statistics:
> |Dataset|$\mu_{scale}^*$|$\sigma_{scale}^*$|
> |--|:--:|:--:|
> |Lyft|0.9428|0.2772|
>
> To confirm, we selected all of our heuristics and hyperparameters on the Lyft dataset, and directly transferred the method into the Ithaca365 dataset *without changing the values*, suggesting at least some generalizability in the method.
>
> > Goal ambiguity from the introduction text
>
> We thank the reviewer for pointing out this possible cause of confusion, and will ensure that our final wording is clearer in our objectives.
>
> [1] BING: Binarized normed gradients for objectness estimation at 300fps
>
> [2] Survey and Performance Analysis of Deep Learning Based Object Detection in Challenging Environments

---

> > ### Comment · Reviewer_Baoq · 2023-08-21
> > **Still positive**
> >
> > Dear all,
> >
> > After going through whole post review text, I remain positive for this submission.

---

### Official Review · Reviewer_vhou · 2023-06-26

**Soundness:** 3 good
**Presentation:** 4 excellent
**Contribution:** 3 good
**Rating:** 6
**Confidence:** 4

**Summary:**

The proposed DRIFT framework is an approach to realize object discovery without labels. DRIFT first extracts foreground proposals based on the PP-score method, and then leverages common-sense heuristics including shape prior, box alignment, and background point filtering, to reward proposed boxes. Reinforcement-learning-based optimization is adopted, maximizing the reward function in a local space. The experimental results demonstrate the superiority of DRIFT over prior self-training methods in terms of efficiency and generalizability.

**Strengths:**

- The framework for 3D object discovery that simplifies common-sense heuristics as Gaussian-based reward signals to fine-tune the detector is interesting, straightforward, and relatively novel. Since the object discovery paradigm does not incorporate pre-defined class labels, it is promising in out-of-domain object perception tasks.
- Experiments are basically comprehensive, with ablations on key reward components and plenty of hyperparameters.
- The paper is well-written, and the ideas are presented clearly, making it easy to follow the authors' arguments and understand their contributions.

**Weaknesses:**

- The authors mainly compare DRIFT with proposal-based (PP-score) baselines.
    - A missing related work for comparison: [a]. [b] is a concurrent work, but is valuable to discuss it though a direct comparison is not feasible.
    - The PP-score method requires multiple traversals to get a superior performance which could limit the generalizability in real-world applications. This helps DRIFT to achieve a closer out-of-domain supervised detector trained on KITTI in the Ithaca365 dataset. But in normal datasets without many traversals, DRIFT actually has a large gap compared to the out-of-domain supervised detector.
    - Current heuristics behave poorly in pedestrians, as shown in the per-class BEV mAP of the pedestrian class in Fig. 6 & Supp-Fig. 1. Heuristics need careful designs and more sophisticated ones are probably necessary to achieve satisfactory results on all categories of objects.
- The reviewer is curious about the training efficiency. Any more insights about why the proposed DRIFT converges much faster than self-training and MODEST?
- Any more insights about the large StD. in Table 1?
- Misc: Fig. 1 is not referenced in the main body.

> [a] Motion Inspired Unsupervised Perception and Prediction in Autonomous Driving. ECCV 2022.
>
> [b] Towards Unsupervised Object Detection from LiDAR Point Clouds. CVPR 2023.

**Questions:**

Please see weaknesses above.

**Limitations:**

The authors have discussed limitations at the end of the paper. The discussion is valuable.

---

> ### Author Rebuttal · Authors · 2023-08-09
>
> Thank you for your positive reviews and thoughtful feedback. We address the individual points below:
>
> >Missing related works
>
> Thank you for pointing to these papers. Unfortunately we did not find code for them, and thus could not compare with them during our rebuttal. We will instead include discussion on them in our final version.
>
> > Generalizability in real-world applications
>
> We would like to highlight that in the real world, it is common for people to follow the same routine, or for different people to share the same route that can be pooled together to produce the multiple traversals. Therefore, while not all publicly available datasets contain multiple traversals, it is highly likely that they become accessible in large quantities when developing self-driving cars in practical contexts.
>
> > Performance on Pedestrians
>
> The limited performance on pedestrians can be mainly attributed to two reasons: First of all, DRIFT itself does not predict classes for its detections, and we could only assign class labels during post-processing based on the most similar shape priors. Sometimes pedestrians are mislabeled as cyclists and vice versa, as they often have similar box sizes. Secondly, the classes are highly imbalanced in both datasets. For example, the Lyft dataset contains 92.47% cars and 5.91% pedestrians. This limits a detector’s ability to learn to detect the smaller classes. We leave this study to a future work.
>
> > Training efficiency
>
> One potential reason is that DRIFT guides boxes to adhere to common sense heuristics using the reward function throughout the entire training process, while self-training and MODEST only leverage them for seed label generation and (for MODEST) filtering between self-training episodes. Therefore, DRIFT more effectively harnesses the heuristics, and leads the detector to learn to predict appropriate boxes much faster.  For additional details, see response to reviewer dARK, Similarity and advantages over self-training methods.
>
> > Large Standard Deviation in Supp. Table 1
>
> The large StD.s can be caused by several reasons: First of all, there are inherent size variations among objects within the same class. For example, a compact car and a SUV both belong to the Car class but have distinct dimensions. Additionally, values in Supp. Table 1 are computed from the box shapes in the Lyft dataset, and there are variations in human annotations.
>
> > Missing reference for Supp. Figure 1
>
> Thanks for pointing this out! We will add a reference for it in our final version.

---

> > ### Comment · Reviewer_vhou · 2023-08-14
> > **Post-rebuttal Comment**
> >
> > Thanks for the feedback.
> >
> > **[Re: Large Std.]** My intent was to ask the large Std. in Table 3 compared with other models in the main paper. The number in my original question was a typo. Sorry for the mistake. Do you have more insights about the large StD. in Table 3?

---

> > > ### Author Response · Authors · 2023-08-15
> > > **Insights into Table 3 StD.**
> > >
> > > In general, bad boxes consistently yield low rewards, as seen in the low mean and low StD. of the "rand boxes" row in Table 3. Good boxes tend to get high rewards on average, but the rewards may vary across different boxes because the signal used to compute the rewards can be noisy though still correlated with box quality. For instance, one part of our reward function, dynamic/static point counts, varies depending on the proximity of the box to the LiDAR sensor, since LiDAR point density decreases with range. As another example, the P2-score we used as an approximate foreground segmentation is a noisy heuristic for the dynamic points. Thus while the rewards are correlated with actual box quality (as shown by the high mean reward for the highest quality boxes, i.e., the ground truth), the correlation is not perfect. That said, note that this large StD. has a minimal impact on DRIFT, as the top *k*% filtering retains mostly good boxes, allowing the detector to learn to identify dynamic objects from them.

---

### Official Review · Reviewer_dARk · 2023-07-03

**Soundness:** 2 fair
**Presentation:** 3 good
**Contribution:** 2 fair
**Rating:** 4
**Confidence:** 4

**Summary:**

This paper proposes a new reinforcement-learning-based framework for unsupervised 3D object detection that uses these common-sense heuristics directly as a reward signal. Avoid handcrafting training examples for each object detector. Furthermore, under the premise of greatly accelerating the convergence speed of the model, this method improves the experimental effect.

**Strengths:**

1. The proposed reinforcement learning-based framework without encoding heuristics into differentiable loss functions and avoids the need to hand-engineer training paradigm.
2. The method proposed by the author has a fast convergence speed and a great algorithm implementation value.
3. The experiments in this article are relatively sufficient, and the charts are easy for readers to understand.

**Weaknesses:**

1. The content arrangement of this article does not seem to be very convenient to read and there are some careless quoting errors that refer to Questions.
2. The author should provide further explanation or derivation in the formula part of the article.

**Questions:**

1. In line 40, can it be understood that the method proposed in this article is to perform finetune on the pre-trained MODEST through a reward-based method? If the answer is yes, is the time comparison in Figure 1 meaningful or fair? Moreover, LLM is mentioned more than once in the article, but this article does not seem to use a specific LLM model. I think this may cause some misunderstandings for readers at the beginning.
2. The method proposed in this paper is more like a specific RAFT method in unsupervised object detection, and the novelty may be limited.
3. The training steps mentioned in Line 56. As far as I know, many self-training methods can achieve the same goal, please tell us your advantages.
4. Lines 126 to 132 in the article seem to give the boxes from the detector a priori information or heuristic constraints from the real world. You are modeling here by a mixture of Gaussian distributions, how to determine mixture weights? In line 115 section. What happens if you use a mixture factor greater than 4? It seems that the model only focuses on objects of these sizes. In line 131, what happens if you don't use the scale operation on it? please give an example.
5. Is Figure 3 drawn from a real dataset? If so, please describe which data were used in the supplementary material. Can this be understood as prior information derived from daily data?
6. The paragraph starting on line 140 means that by scaling, more points are included in the box, this seems to serve the same purpose as jittering mentioned in Sec 3.2 or Intro. What is the difference between them?
7. The sensitivity analysis of mu_{scale} in line 156 should be Tab5.
8. The ablation experiment of Tab4 should complement the other 3 groups and did not analyze the reason for this phenomenon, especially when the effect gap is huge. Because I want to know how the three components of the proposed Reward function affect the detection performance respectively
9. Please confirm the experimental code for reproducing the MODEST in Tables 1 and 2. Whether there is a P2 filtering result seems to be far from the original MODEST paper.


**Limitations:**

One limitation is that this paper uses MODEST as the baseline. If the number of timestamps is not large, continuing to use the p2 score may aggravate the impact of noise on the model. This is a direction that needs to be addressed. Another limitation is that since it is unsupervised 3D object detection, not only static mobile objects should be detected, but all static objects should be detected, which may be beneficial to downstream tasks.

---

> ### Author Rebuttal · Authors · 2023-08-09
>
> We thank the reviewer for their valuable comments, and will incorporate all of the edits/analysis that they suggest. Regarding the mention of LLMs, our method draws inspiration across many different fields (Reinforcement Learning, Object discovery, RLHF) and we had hoped to showcase an example of where such a bridge in these different fields has proven successful. We thank the reviewer for pointing out this potential point of confusion, and will clarify it in the final revision.
>
> > Finetune on the pre-trained MODEST clarification
>
> Our method is not fine-tuning a MODEST detector, and the time comparison in Fig. 1 is correct. The x-axis refers to the number of training epochs *from scratch* needed to obtain the values. The model we are fine-tuning from is pre-trained on noisy seed labels produced by DBSCAN clustering on spatial and PP-score (L203-206), plotted as the yellow dot in Fig. 1.
>
> > Novelty may be limited: similarity to RAFT
>
> This work follows along a broader branch of works commonly known to the Reinforcement Learning (RL) and Controls community as Filtered/Top-K behavior cloning [1]. However, while this field is *well studied for controls applications*, we consider our work a pioneering effort in bringing its advantages into object discovery. The work RAFT [2] also cites RL as inspiration for improving generative modeling. However, the techniques of RL have never been used for object discovery; we demonstrate that by leveraging its tools to optimize non-continuous and non-differentiable rewards, we can obtain incredible performance (tables 1, 2). In summary: 1) we explore the task of object discovery as compared to generative modeling, 2) we formalize novel reward and exploration functions for the setting, which has never been studied before. We additionally hope to point out that, at the time of the submission, the work RAFT had not been published at an official conference venue, and should be considered a concurrent work to ours.
>
> > Question regarding size prior mixture of Gaussian distribution
>
> We comment about shape templates and their generalizability in the general response, and additionally address it here. The mixture weights that we use are computed such that the probabilities at the Gaussian means are equal for stability reasons (L131). We show that the method is robust to mixture weights (Tab. 8) as well as number of mixture factors (general response).
>
> > Reward component analysis
>
> We analyze the components of the reward function in Tab. 4, Fig. 5, and provide additional discussion into the findings. Removing the filtering step causes the object detector to be unable to distinguish between foreground objects and static background, thus the drop in performance (Tab. 4, L3). To better understand the reward ablation, we visualize the resulting predictions in Fig. 5 of the main paper. Removing the size reward causes the model to place bounding boxes of incorrect shape (Fig. 5, lower left). Removing the alignment reward results in boxes that have points in the center of the box, optimizing for large boxes that capture all the dynamic objects (Fig. 5, lower right). Because these failure cases are size and shape specific, the metric of mAP at IoU 0.7 and IoU 0.5 cannot capture these kinds of errors, thus the large drop in perceived performance (Tab. 4, L2&4).
>
> > Clarifications about Figure 3
>
> Figure 3 is plotted from real data, specifically the Lyft L5 object detection dataset. We will include this in the final version, and thank the reviewer for pointing out this error.
>
> > Clarification about scale verses jittering
>
> To clarify, jittering is exploration of the box proposals to search for potentially better candidates, while scale is referring to the ratio near a particular box used to calculate the reward associated to the box. All boxes that were produced from jittering considers their own o(b) set of points when computing the reward (and thus, the scale), and the best boxes from all jittered boxes are used to update the model (L139-140, 170-172).
>
> > Clarification about baseline reproduction
>
> We report the results on mAP at IoU 0.5 and 0.7, which is different from the results reported in MODEST (IoU 0.25, 0.5). We made this choice to follow the precedent set in prior object detection works, KITTI dataset and Lyft dataset. We used the official implementation for baseline reproduction. We will provide results for IoU 0.25 in the final submission, but note that at such a low IoU the metric captures only the localization accuracy of the model. Our work is able to localize objects as well as predict the correct shape completion, as shown in the higher performance at the more strict IoU matches (Tab. 1).
>
> > Clarification regarding number of timestamps and impact of noise
>
> We hope to clarify that the computation uses repeated traversals as opposed to consecutive timestamps to compute P2-score, and is actually more robust to noise per-sequence in the current traversal as demonstrated by [3, 4].
>
> > Limitation: discovery of static objects
>
> We agree with the reviewer that this is a limitation of the current method (L304-305); however, we are limited by the data and labels that are present in an academic setting. In this work, we focus on object discovery for self-driving following prior self-driving object detection tasks. We note that there are only labels for dynamic objects, which is necessary for evaluation. In theory, however, one can define a reward function for any objects they desire to discover, both static or dynamic, and leave this to future works (L307).
>
> [1] Kumar et al. 2022. When Should We Prefer Offline Reinforcement Learning Over Behavioral Cloning?
>
> [2] Dong et al. 2023. RAFT: Reward rAnked FineTuning for Generative Foundation Model Alignment (in submission)
>
> [3] Barnes et al., 2017. Driven to Distraction: Self-Sup. Distractor Learning for Robust Monocular Visual Odometry
>
> [4] You et al., 2022. Hindsight is 20/20: Leveraging Past Traversals to Aid 3D Perception

---

> > ### Comment · Reviewer_dARk · 2023-08-19
> >
> > I appreciate the efforts the authors spent on their rebuttal! It solves some of my concerns. But I have two other questions.
> >
> > 1. As mentioned by reviewer iC9C, 'Since the approach is supposed to find objects unsupervised, hyperparameter tuning and then manually checking the results by a human who looks at the bounding box detections in the point cloud may improve the results but not in a fair way. It goes against the idea of unsupervised object detection.' While your answer is that these parameters are very robust to other datasets, this seems to be at the expense of learning from the data, can you give another explanation?
> >
> > 2. The scoring function designed in Sec3.1 seems too heuristic. In 2d unsupervised object detection, there are some works like[1] which leverage VLMs to generate or filter pseudo labels, can you refer to these ideas to further improve the effect of the model or reduce the complexity of the model?
> >
> > But in my opinion, I think the novelty of this paper is still limited, and I insist on my points for now.
> >
> > [1] Zhao, Shiyu, et al. "Exploiting unlabeled data with vision and language models for object detection." European Conference on Computer Vision. Cham: Springer Nature Switzerland, 2022.

---

> > > ### Author Response · Authors · 2023-08-20
> > > **Thank you for your response.**
> > >
> > > We respectfully disagree that this method lacks novelty; it’s a compellingly simple, effective, and robust framework, as shown in the variety of ablations both in the paper and in the subsequent rebuttal. However, we agree that there are *ways that the method can be expanded* within this framework. Part of our contribution lies in the general framework that can leverage functions of any type, differentiable or not, in an efficient manner, and bridges the concept of exploration mechanism, common for the field of RL, successfully into object discovery.
> > >
> > > We thank the reviewer for pointing to another way we can extend the reward and exploration function to incorporate additional modes of information such as language [1]. While out of scope for this work, our future works aim to explore reward/exploration functions that are appropriate for such settings (L307-308). However, our current contribution re-thinks the problem of object discovery in a fundamentally new way, and in doing so, yields incredible performance gains. Indeed, reviewers iC9C, vhou, and Baoq consider the method to be novel and a valuable contribution, and we would be happy if reviewer dARk can reconsider their review under this light.
> > >
> > > > Hyperparameter tuning, domain knowledge, and learning from data.
> > >
> > > We base the selection of many of our hyperparameters on human domain knowledge, which does not necessarily need direct learning from the data. For instance, we can readily obtain dimensions of transportation tools from online sources [2], and estimate human body shapes using health statistics reports from the CDC [3]. This strategic application of domain knowledge enables us to scale to large amounts of scenes and does not require human involvement for each individual scene. We contend that to do unsupervised learning, we must assume some knowledge, either from domain knowledge, learned from human labels, or distilled from other datasets. In this work, we assumed access to simple and easily accessible domain knowledge and believe it is the most scalable solution.
> > >
> > > > Leveraging VLMs to generate or filter pseudo labels
> > >
> > > To clarify, the scoring function is the objective we wish to optimize. Proposals (i.e., pseudo-labels) are guided by exploration to encourage identifying boxes that can improve the scores (i.e., reward) that are obtained. Better proposals will help get the model to better regions quicker, thus increasing the efficiency of the method, but ultimately the objective of the method is to maximize the reward. Under our framework, if the aim is to improve the reward function, one can use VLMs to lift features from 2D scene images, then associate it with another model to the 3D points. The final reward can be some function that encourages detections around features that correspond to VLM features. This would be the most direct analogy to how the work [1] utilized the VLM features. However, this would require a 2D-to-3D model, since all VLMs are currently trained under the 2D image domain, and at this moment, none exist for 3D data. In this way, our current reward formulation would be the most simple and straightforward, but we are incredibly encouraged if such a potential signal should, in the future, become available. We thank the reviewer for pointing out this interesting work and will expand on discussion regarding it.
> > >
> > > [1] Zhao, Shiyu, et al. "Exploiting unlabeled data with vision and language models for object detection." European Conference on Computer Vision. Cham: Springer Nature Switzerland, 2022.
> > >
> > > [2] https://www.dimensions.com/classifications/transport
> > >
> > > [3] Fryar CD, Kruszon-Moran D, Gu Q, Ogden CL. Mean Body Weight, Height, Waist Circumference, and Body Mass Index Among Adults: United States, 1999-2000 Through 2015-2016. Natl Health Stat Report. 2018 Dec;(122):1-16. PMID: 30707668.

---

### Official Review · Reviewer_ZfQG · 2023-07-04

**Soundness:** 3 good
**Presentation:** 3 good
**Contribution:** 3 good
**Rating:** 6
**Confidence:** 4

**Summary:**

This paper proposes DRIFT, a novel reward fine-tuning method for unsupervised object discovery with point cloud input. Specifically, three reward methods are proposed to identify good bounding boxes. First, shape prior reward prefers bounding boxes with similar sizes to the prototypes.  Second, an alignment reward gives high scores to boxes that have most of the LiDAR points near the box edge. Third, a filter reward follows the spirit that a good box should contain more dynamic points than background points. By using these reward methods for refining the object detector, DRIFT achieves more accurate object discovery, and it also converges much faster than the previous method.

**Strengths:**

(1) The proposed reward fine-tuning method is well-motivated and improves object discovery accuracy compared to the previous work.

(2) In addition to improving the detection performance, the proposed method can also greatly improve the training speed.

(3) The expression of the paper is clear, and the figures are intuitive.

**Weaknesses:**

(1) Although intuitive, I think the prior that most points fall on the lateral surfaces of an object is too absolute, e.g., a lot of LiDAR points will fall on the hood and front window of the oncoming vehicles beside lateral surfaces. Apart from this, using Gaussian distribution as an approximation is also not accurate, because few points fall out of the box edge, as shown in Figure 3.

(2) I think a missing ablation is the contribution of the exploration strategy to the final detection performance, similar to the ablation on the reward methods in Table 4.

**Questions:**

(1) Authors should provide complete experiment results and refine their experimental analysis correspondingly, as some of the experiments are still running at the time of submission (L234 & L258).

(2) In L224, the authors say they include evaluation results with IoU at 0.25 in the supplementary, but I don't see these results. Authors should add these results and explain why they don't provide these results in the main paper.

**Limitations:**

The authors provide limitations.

---

> ### Author Rebuttal · Authors · 2023-08-09
>
> > About using the Gaussian distribution as an approximation
>
> We agree with your point that the Gaussian distribution approximation might not be the most accurate one, but from our observation the hood or front window of vehicles only contribute to a small fraction of the point cloud (Figure 2 in the main paper shows an example) since lidar reflections are sparser on glasses or surfaces parallel to the beam. We also contend that this approximation is one of the simplest, robust approximations while yielding satisfactory performance. We showed ablation study in Table 5 in the main paper, Table 2 and 8 in the supplementary material about robust results of varying the Gaussian parameters. We hope this could serve as a first but simple and strong step, and encourage further research.
>
> > About the contribution of exploration strategy:
>
> We note that we have included Table 6, 7, 8 in the main paper, Table 4, 5, 6 in the supplementary material for contributions of different exploration strategies. In addition to Table 6 in the main paper, we note that if we do not use any exploration at all, the detection performance will drop to 0 (we will add an additional line in Table 6 for this in the final version) due to confirmation bias (which is a key problem with self-training), in which case the model gets indulged in is own proposals and the training loss cannot provide meaningful gradients.
>
> > Final number
>
> We report our results for up to 300 epochs below:
>
> |                 | IoU 0.5 | IoU 0.7 |
> |-----------------|:-------:|:-------:|
> | No Finetuning   |   23.9  |   10.5  |
> | MODEST (600 ep) |   39.6  |   18.8  |
> | DRIFT (30ep)    |   38.3  |   23.1  |
> | DRIFT (60 ep)   |   41.8  |   26.7  |
> | DRIFT (120 ep)  |   45.3  |   29.6  |
> | DRIFT (180 ep)  |   43.2  |   30.2  |
> | DRIFT (300 ep)  |   44.6  |   31.3  |
>
> We observed that the method converged soon after the final reported numbers at 120 epoch. We will include the final number into the report.
>
> > About results for IoU 0.25:
>
> Thanks for pointing this out! We accidentally missed this in the supplementary material, and we provide them below. We will add this to the final version.
>
> Table Results for IoU 0.25
>
> |                      | 0-30 | 30-50 | 50-80 | 0-80 |
> |----------------------|:----:|:-----:|:-----:|:----:|
> | No Finetuning        | 63.5 |  34.9 |  6.0  | 37.5 |
> | Self Train. (60 ep)  | 67.7 |  43.2 |  8.7  | 43.1 |
> | Self Train. (600 ep) | 67.7 |  48.0 |  13.3 | 45.5 |
> | MODEST (60 ep)       | 68.5 |  46.2 |  10.5 | 45.1 |
> | MODEST (600 ep)      | 73.6 |  56.8 |  21.0 | 53.6 |
> | DRIFT (60ep)         | 72.3 |  51.5 |  19.2 | 50.7 |
> | DRIFT (120 ep)       | 72.5 |  51.7 |  25.8 | 52.9 |
> | Supervised on KITTI  | 78.6 |  53.9 |  26.1 | 55.3 |
> | Supervised on Lyft   | 81.8 |  63.6 |  40.0 | 64.2 |
>
> One thing to bear in mind is that IoU 0.25 metric is more evaluating “localization”, i.e. if there is a bounding box with a tiny overlap with the GT. However, our method excels at the size and proper orientation, which is better captured at higher IoUs metrics. Basically, MODEST can put a box, but it’s not able to figure out the size, but our method does both. As stated in the main paper, we use 0.5 and 0.7 for 1) following the KITTI and Lyft standards of reporting, 2) emphasizing the strength of our method.

---

### Official Review · Reviewer_iC9C · 2023-07-06

**Soundness:** 3 good
**Presentation:** 4 excellent
**Contribution:** 3 good
**Rating:** 6
**Confidence:** 3

**Summary:**

This work contributes to unsupervised object discovery in LIDAR point clouds. It defines a few typical properties of customary bounding boxes in LIDAR point clouds and then develops Rewards for a Reinforcement Learning algorithm to learn this, lacking a gradient for direct optimization.

It is based on Persistency Prior scores defined by the MODEST approach but adds several steps to improve bounding box selection. Furthermore they ablate many factors, e.g. the impact of enforcing a certain shape on bounding boxes. While they can only compare on a limited amount of epochs due to time constraints, possibly due to the plenty ablation studies, they show they outperform the existing approach in that domain.


**Strengths:**

The paper improves performance by a large margin in a very novel research area. Even if the approach itself may have limited use, it is straightforward to see the value e.g. in combination with a pre-trained traffic participant classifier.

The easy and intuitive explanation of the reward shaping is very clear. Figure 2 helps to get a very quick insight into the idea. The whole paper does a good job at combining intuitive explanations with mathematical formulations.

The approach has many individual steps and the pseudo-algorithm helps understanding.

The used Lyft Level 5 Perception dataset is a good choice for comparing with the state of the art.

The ablation studies are suitable to show that all used steps together achieve the performance.

All figure captions guide the reader nicely towards the main message that figure should convey.

**Weaknesses:**

It seems this work takes many ideas from MODEST so the idea of using an object detector and defining common sense properties is not novel to this work. It needs to be read with that contribution offset in mind.

The title is maybe a bit boasting. Teaching cars to see sounds like solving most if not all perceptual problems while this approach, coming from unsupervised object detection, only separates moving objects from background.

Having still a bit to train is not optimal but understandable. The paper feels a bit rough around the edges, the DRIFT abbreviation is explained twice.

The approach is a bit complex with many parameters to tune. Even though the ablation studies show the value of all individual components, it is hard to judge how much hyperparameter optimization vs. unsupervised modeling performance is present.

**Questions:**

2 Related Works: 3D Object Detection: I would disagree it is ideal to train detectors in an unsupervised way. Maybe ideal could be a combination to achieve best performance at reasonable costs while being able to assign actual labels (car, pedestrian, etc.). Maybe a statement like that could be just removed.

How robust is the algorithm to the amount of traversals? This is not touched at all unless I overlooked it.

Jittered boxes and then NMS has strong similarity to region proposals in approaches like Mask R-CNN. Maybe it could be good comparing this historic approach in the Related Work.

What is the impact of the quality of the first pre-trained detector? If there is a false negative detection, how can this algorithm improve a non-existing bounding box?

In Reward Finetuning for Model Alignment, what are "human values"? Do the models output unexpected or wrong output or are there ethical considerations? The term seems confusing.

How have the lambda factors and mu_scale and sigma_scale been found? Similarly how was the scale of the scaled up box selected when designing the Alignment Reward? Since the approach is supposed to find objects unsupervised, hyperparameter tuning and then manually checking the results by a human who looks at the bounding box detections in the point cloud may improve the results but not in a fair way. It goes against the idea of unsupervised object detection.

In 3.2 what is meant by "In effect, this encourages all non-persistent points ... to propose boxes". How do points propose boxes? This is not like Yolo where each grid cell is associated with a fixed number of region proposals right?

Couldn't this approach potentially be expanded to actually label Car, Pedestrian, Cyclist and Truck based on the shape priors?

It would be worthwhile investigating the impact of false negatives to understand the impact of the approach vs. the failure of the first step of bounding box proposal. The authors did compare different ways of box predictions but that is different from quantifying the performance. Knowing how many false negatives are produced by the first step would already give some insight into this. Or is that the No Finetuning case?


**Limitations:**

There is only one comparable work which is an unavoidable limit. The authors solved this by also comparing against other tasks, i.e. the supervised case.

Due to the deadline DRIFT was not trained to convergence, however it seems to outperform it's competitor already. For the camera ready final numbers should be entered.

In their limitations the authors address most of the concerns in Weaknesses.

---

> ### Author Rebuttal · Authors · 2023-08-09
>
> Thank you for your feedback! We address the individual points:
>
> > Novelty: compared to MODEST
>
> We highlight that although MODEST and DRIFT both use commonsense properties, MODEST uses them only when generating seed labels and filtering between self-training rounds. In contrast, DRIFT directly incorporates common sense rules throughout the training process using the reward function, which is a significantly more efficient way of leveraging the knowledge. In addition, DRIFT takes a fundamentally different approach towards object discovery by leveraging imitation learning tools from Reinforcement Learning that are designed to handle the non-continuous, non-differentiable nature of the commonsense properties. In doing so, DRIFT displays significantly faster convergence and stronger performance than MODEST (Fig. 1).
>
> > Title and writing
>
> We will adjust our title to make it more appropriate and descriptive of our task. We will remove the statement that it is ideal to train detectors in an unsupervised way and the repetitive explanation of the abbreviation.
>
> > Robustness to the amount of traversals
>
> The Lyft dataset has around 5 repeated traversals per location, and Ithaca365 has around 20 traversals per location. DRIFT performs competitively on both datasets, indicating that it is robust to the amount of traversals, and the number of traversals needed does not have to be very large.
>
> > Comparison with Mask R-CNN
>
> Thank you for pointing this out; we will include corresponding discussion in our final version.
>
> > What is the impact of the quality of the first pre-trained detector?
>
> The pretrained detector can have bad quality (e.g. Tab. 1 “no finetuning”), but it needs to have some functionality and cannot be completely randomly initialized. A pretrained detector with better quality may further improve DRIFT’s performance.
>
> > ​​In Reward Finetuning for Model Alignment, what are "human values"?
>
> The meaning of “human values” is context dependent. It could refer to output quality such as accuracy or coherence (e.g. [41] in Related Works) or ethical considerations (e.g. [30] in Related Works), etc. In the context of our task it refers to the common sense rules that the boxes should follow. We will make this clearer in our final version.
>
> > How have the [hyperparameters] been found?
>
> The lambda factors were found by hyperparameter tuning on the Lyft dataset. Intuitively, the selected values aim to balance each component of the reward and prevent one from dominating the others. The mu_scale, sigma_scale and the x2 scaled up range were adopted based on common sense reasoning: The points should in general fall close to the inner surface of the bounding box, leading the mu_scale to be slightly smaller than 1. And we intuitively consider x2 scale a reasonable range to encompass points associated with the object a box aims to detect. Our ablation in Supp. Table 2 further indicates that reasonable alternative choices of mu_scale would yield similar performance. We additionally ablate the standard deviation sigma_scale, and observe that it is not sensitive to the choice of value:
>
> | Scale St. Dev. | IoU 0.5 | IoU 0.7 |
> |----------------|:-------:|:-------:|
> |   0.1 St. Dev. |   34.2  |   20.9  |
> |   0.2 St. Dev. |   38.3  |   23.1  |
> |   0.3 St. Dev. |   38.1  |   20.0  |
>
> We would also like to highlight that we directly applied the hyperparameters we identified on the Lyft dataset to Ithaca365. Despite the differences in environment and data distribution, DRIFT attains strong performance on Ithaca365 using the hyperparameters from Lyft. This indicates that our default hyperparameters are very generalizable, and hyperparameter tuning or other human involvement may not be necessary when training DRIFT for a new domain.
>
> > In 3.2 what is meant by "In effect, this encourages all non-persistent points ... to propose boxes"?
>
> This is referring to the first stage of PointRCNN, which classifies each point as either foreground or background, and generates box proposals from foreground points. A focal loss is applied to the foreground/background classification. Here, we modify the classification labels used for the loss, such that a non-persistent (low PP-score) point predicted as background is still treated as foreground. A similar approach has been adopted in [1]. This modification encourages the detector to generate proposals around non-persistent points.
>
> > Labeling based on the shape priors
>
> Yes, DRIFT has the potential to provide a rough class labeling based on its shape priors, e.g. like what we did in the “Extension to Detection with Classes” section. It is worth noting that some classes (e.g. pedestrians vs. cyclists) can have similar sizes, and thus cannot be completely separated based on shape. Additional heuristics may be needed to improve the accuracy of the labeling.
>
> > False negatives
>
> As the detector gets trained for longer, it becomes better at identifying dynamic objects and is less likely to have false negatives. We report the recall at various epochs below. We also empirically find that additionally sampling boxes around non-persistent points (as a way to capture potential false negatives) does not improve the end performance (Table 6).
>
> | Recall @ Ep. | IoU 0.5 |
> |--|:--:|
> | 30 | 0.47 |
> | 60 | 0.51 |
> | 90 | 0.53 |
> | 120 | 0.56 |
>
> > Final numbers
>
> We provide the results below, and include discussion in ZfQG's response.
>
> |  | IoU 0.5 | IoU 0.7 |
> |--|:--:|:--:|
> | No Finetuning   |   23.9  |   10.5  |
> | DRIFT (300 ep)  |   44.6  |   31.3  |
>
> [1] You, Yurong, et al. 2022. Unsupervised Adaptation from Repeated Traversals for Autonomous Driving.

---

> > ### Comment · Reviewer_iC9C · 2023-08-14
> >
> > Read the comments of the authors and other reviewers comments. I am content with the responses.

---

### Author Rebuttal · Authors · 2023-08-09

We express gratitude to the reviewers for their constructive feedback on our work and appreciate their acknowledgment that the writing is "well written" and "easy and intuitive" to follow [iC9C, ZfQG, vhou, Baoq]. To reiterate, our work introduces a novel adaptation of Reinforcement Learning (RL)-based methods for unsupervised object discovery from LiDAR points, which surpasses prior works in both accuracy and training efficiency. We appreciate the reviewers' recognition that "[our method] improves performance by a large margin in a very novel research area" [iC9C] and that our method is noted to "significantly improve over [prior works]" by all reviewers. Reviewer dARk concisely characterizes our work as a reinforcement learning-based framework that mitigates the need to encode heuristics into differentiable loss functions. In summary, we are thankful that reviewers have found our work to be "straightforward" [iC9C, vhou] and "well-motivated" [ZfQG, Baoq].

Additionally, we present a general comment regarding the advantages over self-training methods:

Our method, DRIFT, significantly outperforms vanilla self-training for object discovery and converges at a much faster rate (Table 1, 2, Figure 1: Self-Train baseline). Traditional self-training iteratively generates pseudo-labels and retrains the model, requiring convergence before generating the next set of pseudo-labels. In unsupervised scenarios, training a detector to mimic pseudo-labels from a model lacking ground truth supervision can lead to undesirable artifacts, further amplified by repeated training (confirmation bias). Our method mitigates the problem by drawing on the field of RL: the exploration component is crucial for our method (shown below), and is not present in traditional self-training methods. By performing local exploration instead of simply updating from its own predictions, DRIFT avoids confirmation bias and ensures that labels improve over what it predicts. Thus, DRIFT is able to perform updates per-training iteration as opposed to per self-training round (which is generally 60 epochs, times number of iterations) which allows it to converge significantly faster and achieve higher performance.

| | IoU 0.5 | IoU 0.7 |
|--------------|:-------:|:-------:|
| No Exploration       |   0.0   |   0.0   |
| W/ Exploration       |   41.8  |   26.7  |


We also provide a discussion on the shape templates and their generalizability, brought up by reviewers dARk and Baoq:

The setting we’re studying assumes domain knowledge over object sizes for which we wish to discover, and we follow prior self-driving works using the classes: cars, pedestrians, cyclists, and trucks (Supplementary, Tab. 1). Our main results use the ground truth mean and variance class sizes computed on the Lyft dataset, and we show that it can generalize to the Ithaca365 dataset *without changing the values* (Table 2). We also ablate the mixture weights in the Supplementary and show that the results are not sensitive to the values chosen (Table 8):

| Shape St. Dev.   | IoU 0.5 | IoU 0.7 |
|------------------|---------|---------|
| 0.5 * I St. Dev. |    34.5 |   21.3  |
| 0.2 * I St. Dev. |    37.1 |   25.5  |
|    True St. Dev. |    38.3 |   23.1  |

Per reviewer dARk’s suggestion, we further ablate the number of mixture factors (class size priors) greater than 4. We compare to the ground truth number of mixture factors (4) to increasing the number of mixture factors at constant standard deviation and mixture weights:

| Num. Factors | IoU 0.5 | IoU 0.7 |
|--------------|:-------:|:-------:|
|       4 (GT) |   38.3  |   23.1  |
|            5 |   37.0  |   23.0  |
|            6 |   36.0  |   22.7  |

Here, we add in “pseudo-class” size priors that are larger than the largest class at 10 and 15-meter lengths, which do not exist in the data. However, even in such extreme cases we note that our method is robust to the number of mixture factors.

---

### Author Response · Authors · 2023-08-19

Dear reviewers,

 We thank you so much for your time and effort, and would be happy to answer any further questions
 you may have before the discussion period ends. Please let us know if any issues remain
 and/or if there are any additional clarifications we can provide.
 If you are satisfied with our rebuttal, we would appreciate it if you could reconsider your
 score.

Best regards,

Authors

---

### Decision · Program_Chairs · 2023-09-21

**Decision:**

Accept (poster)

**Comment:**

This paper investigates adapting a reward fine-tuning based reinforcement learning method for learning to detect objects from LiDAR points without utilizing training labels. The proposed approach aims to mimic human feedback by combining multiple heuristics into a reward function, instead of using the labels. The manuscript received the following ratings: weak accept,weak accept, borderline reject, weak accept and weak accept. While the reviewers appreciated the motivation, idea and performance of the method, they also raised some concerns. Authors responded to reviewer's queries by submitting a rebuttal. The rebuttal largely addressed the initial concerns raised by the reviewers. Given that four reviewers are generally positive about the paper as well as author's rebuttal, the recommendation is accept. Authors are strongly encouraged to take into account the suggestions of the reviewers as well as changes acknowledged in the rebuttal when preparing the final manuscript.